# ncRNAs-mediated TIMELESS overexpression in lung adenocarcinoma correlates with reduced tumor immune cell infiltration and poor prognosis

**Xinliang Gao**[1], **Mingbo Tang**[1], **Suyan Tian**[2], **Jialin Li**[1], **Shixiong Wei**[1], **Shucheng Hua**[3], **Wei Liu**[1] *

**1** Department of Thoracic Surgery, The First Hospital of Jilin University, Changchun, Jilin Province, PR China, **2** Division of Clinical Research, The First Hospital of Jilin University, Changchun, Jilin Province, PR China, **3** Department of Respiratory Medicine, Center for Pathogen Biology and Infectious Diseases, Changchun, Jilin Province, PR China

* l_w01@jlu.edu.cn

**Data Availability Statement:** The mRNA expression data and LUAD clinical information used in this study are available in TCGA (https://

## Abstract

Lung adenocarcinoma (LUAD) has a poor prognosis. Circadian genes such as *TIMELESS* have been associated with several pathologies, including cancer. The expression of *TIMELESS* and the relationship between *TIMELESS*, infiltration of tumors and prognosis in LUAD requires further investigation. In this study, we investigated the expression of *TIMELESS* and its association with survival across several types of human cancer using data from The Cancer Genome Atlas (TCGA) and the Genotype-Tissue Expression Program. Noncoding RNAs (ncRNAs) regulating overexpression of *TIMELESS* in lung adenocarcinoma (LUAD) were explored with expression, correlation, and survival analyses. Immune cell infiltration and biomarkers were analyzed between different *TIMELESS* expression levels. The relationship between *TIMELESS* expression and immunophenoscores, which were used to predict response to immunotherapy, was evaluated. *TIMELESS* was identified as a potential oncogene in LUAD. NcRNA analysis showed MIR4435-2HG/hsa-miR-1-3p may interact with *TIMELESS* in a competitive endogenous RNA network in LUAD tumor tissues. Most immune cells were significantly decreased in TCGA LUAD tumor tissues with high *TIMELESS* expression except for CD4$^+$T cells and Th2 cells. *TIMELESS* expression in LUAD tumor tissues was significantly negatively correlated with neutrophil biomarkers, dendritic cell biomarkers (*HLA-DPB1*, *HLA-DQB1*, *HLA-DRA*, *HLA-DPA1*, *CD1C*) and an immunophenoscore that predicted outcomes associated with the use of immune checkpoint inhibitors. These findings imply that ncRNAs-mediated *TIMELESS* overexpression in LUAD tumor tissues correlated with poor prognosis, reduced immune cell infiltration in the tumor microenvironment, and poor response to immune checkpoint inhibitors.

## Introduction

In 2020, lung cancer was responsible for approximately 1.8 million deaths, and was the most common cause of cancer-related mortality [1]. Non-small cell lung cancer (NSCLC) is a key

portal.gdc.cancer.gov/) and GEPIA (http://gepia.
cancer-pku.cn/).

**Funding:** The authors received no specific funding
for this work.

**Competing interests:** The authors received no
specific funding for this work.

type of lung cancer. Lung adenocarcinoma (LUAD) is the most common subtype of NSCLC,
and incidence rates have continued to increase in recent years [2]. The application of novel
diagnostic and therapeutic technologies, including immunotherapy [3, 4], has improved prog-
nosis in NSCLC, but the overall five-year survival rate remains low, at 25% according to The
Surveillance, Epidemiology, and End Results (SEER) [5]. There is clinical need for predictive
biomarkers and novel therapeutic strategies in LUAD.

The molecular circadian clock is an endogenous timing system that has been associated
with several pathologies, including cancer [6, 7]. *TIMELESS* is a core circadian gene that con-
trols circadian rhythmicity. *TIMELESS* may be involved in carcinogenesis by modulating cell
cycles, DNA repair, and tumor immunity [8–13]. The expression of *TIMELESS* is increased
and associated with poor patient outcomes in NSCLC [14, 15]. The expression of *TIMELESS*
and the association of *TIMELESS* with tumor immune cell infiltration and prognosis in LUAD
remain to be elucidated.

Competitive endogenous RNAs (ceRNAs), including long non-coding RNAs (lncRNA),
pseudogene transcripts, and circular RNAs (circRNAs), regulate each other as they share
microRNA (miRNA) recognition elements [16]. lncRNAs regulate transcription in mamma-
lian circadian systems [17, 18]. The objectives of this study were to investigate: 1) the expres-
sion of *TIMELESS* and its association with survival across several types of human cancer; 2)
the regulation of *TIMELESS* by non-coding RNAs (ncRNAs) in LUAD; and 3) the relationship
between *TIMELESS* and infiltration of tumors in LUAD to predict outcomes associated with
the use of immune checkpoint inhibitors. Findings indicate that ncRNA mediate upregulation
of *TIMELESS*, which is associated with tumor immune cell infiltration and poor prognosis in
LUAD.

## Materials and methods

### Differential genes expression

In the 33 tumor classes included in The Cancer Genome Atlas (TCGA) datasets, we selected
18 tumor types that contained more than five normal tissue samples (Bladder Urothelial Carci-
noma BLCA, Breast Invasive Carcinoma BRCA, Cholangiocarcinoma CHOL, Colon Adeno-
carcinoma COAD, Esophageal Carcinoma ESCA, Glioblastoma Multiforme GBM, Head and
Neck Squamous Cell Carcinoma HNSC, Kidney Chromophobe KICH, Kidney Renal Clear
Cell Carcinoma KIRC, Kidney Renal Papillary Cell Carcinoma KIRP, Liver Hepatocellular
Carcinoma LIHC, Lung Adenocarcinoma LUAD, Lung Squamous Cell Carcinoma LUSC,
Prostate Adenocarcinoma PRAD, Rectum adenocarcinoma READ, Stomach Adenocarcinoma
STAD, Thyroid Carcinoma THCA, and Uterine Corpus Endometrial Carcinoma UCEC). The
mRNA expression data from these 18 cancers were downloaded from TCGA database. Differ-
ential expression analysis using the Limma package in R (version 4.0.1) identified altered
*TIMELESS* expression between tumors and adjacent normal tissues. The Gene Expression Pro-
filing Interactive Analysis database (GEPIA) (http://gepia.cancer-pku.cn/) [19] was used to
compare and visualize TCGA tumor tissues to normal tissues from TCGA and individuals
without cancer from the Genotype Tissue Expression (GTEx) Program.

### GEPIA database analysis

Package survival in R was used to determine the overall survival (OS) and disease-free survival
(RFS) between different *TIMELESS* expression and lncRNA expression across multiple cancer
types, including LUAD. Two groups of high and low expression were established using the
median value as a cut-off.

## Candidate miRNA and lncRNA prediction

StarBase (sRNA target Base) (http://starbase.sysu.edu.cn/) [20] was used to identify miRNA binding sites in *TIMELESS* and lncRNAs that could bind candidate miRNAs involved in the regulation of *TIMELESS*. miRNA binding sites were predicted by the following programs: PITA, RNA22, miRmap, microT, miRanda, PicTar, and TargetScan. Candidate miRNAs and lncRNAs were identified based on correlations of miRNA expression (spearman |R| >0.2 p<0.05) or lncRNA expression (spearman |R| >0.1 p<0.05) with *TIMELESS* expression (analyzed and visualized by function cor.test and package ggplot2 in R).

## Tumor immune cell infiltration and response to immune checkpoint inhibitors

Single-sample gene set enrichment analysis (ssGSEA) was used to investigate the proportions of 28 types of immune cells in immune cell infiltration in TCGA LUAD tumor tissues [21]. The Cancer Immunome Atlas (TCIA) (https://tcia.at/) was used to predict response to immune checkpoint inhibitors using an immunophenoscore that predicts response to immunotherapy with Cytotoxic T Lymphocyte-Associated Antigen-4 (CTLA-4) and Programmed Death 1(PD-1) blockers [22]. Immunophenoscores were correlated with *TIMELESS* expression using Spearman's correlation.

## Statistical analysis

Statistical analyses were performed using online databases or R software, as described above. p <0.05 was considered statistically significant.

## Results

### *TIMELESS* expression and prognostic utility

Data from TCGA showed *TIMELESS* expression in tumor tissues was significantly increased vs. adjacent normal tissues in 16 types of cancer, and significantly decreased vs. adjacent normal tissues in 2 types of cancer (**Fig 1A**). *TIMELESS* expression was significantly increased in TCGA tumor tissues across 14 types of cancer vs. normal tissues from individuals without cancer in the GTEx Program (**Fig 1B**). Among these 14 tumor tissues, high *TIMELESS* expression in tumor tissues was significantly associated with poor OS in KIRC, LIHC, and LUAD and poor RFS in LIHC and LUAD (**Fig 1C**). These data suggest *TIMELESS* may be utilized as a prognostic marker in LUAD.

### *TIMELESS* upstream miRNAs

A total of 85 miRNAs with potential to interact with *TIMELESS* were identified in TCGA LUAD tumor tissues (**Fig 2A**). According to published literature, there is negative feedback regulation of the target gene by miRNAs; therefore, we expected a negative correlation between candidate miRNA and *TIMELESS* expression and downregulation of candidate miRNAs in LUAD tumor tissues vs. adjacent normal tissues. In TCGA LUAD tumor tissues, *TIMELESS* expression showed a significant negative correlation with hsa-miR-1-3p, hsa-miR-145-5p, hsa-miR-181a-5p, hsa-miR-224-3p and hsa-miR-551a expression (**Table 1**; **Fig 2B**). Only hsa-miR-1-3p was significantly downregulated in LUAD tumor tissues vs. adjacent normal tissues (**Fig 2C**). In the TCGA LUAD cohort, high hsa-miR-1-3p expression in tumor tissues was significantly associated with better OS (**Fig 2D**). These data imply that hsa-miR-1-3p may regulate *TIMELESS* in LUAD.

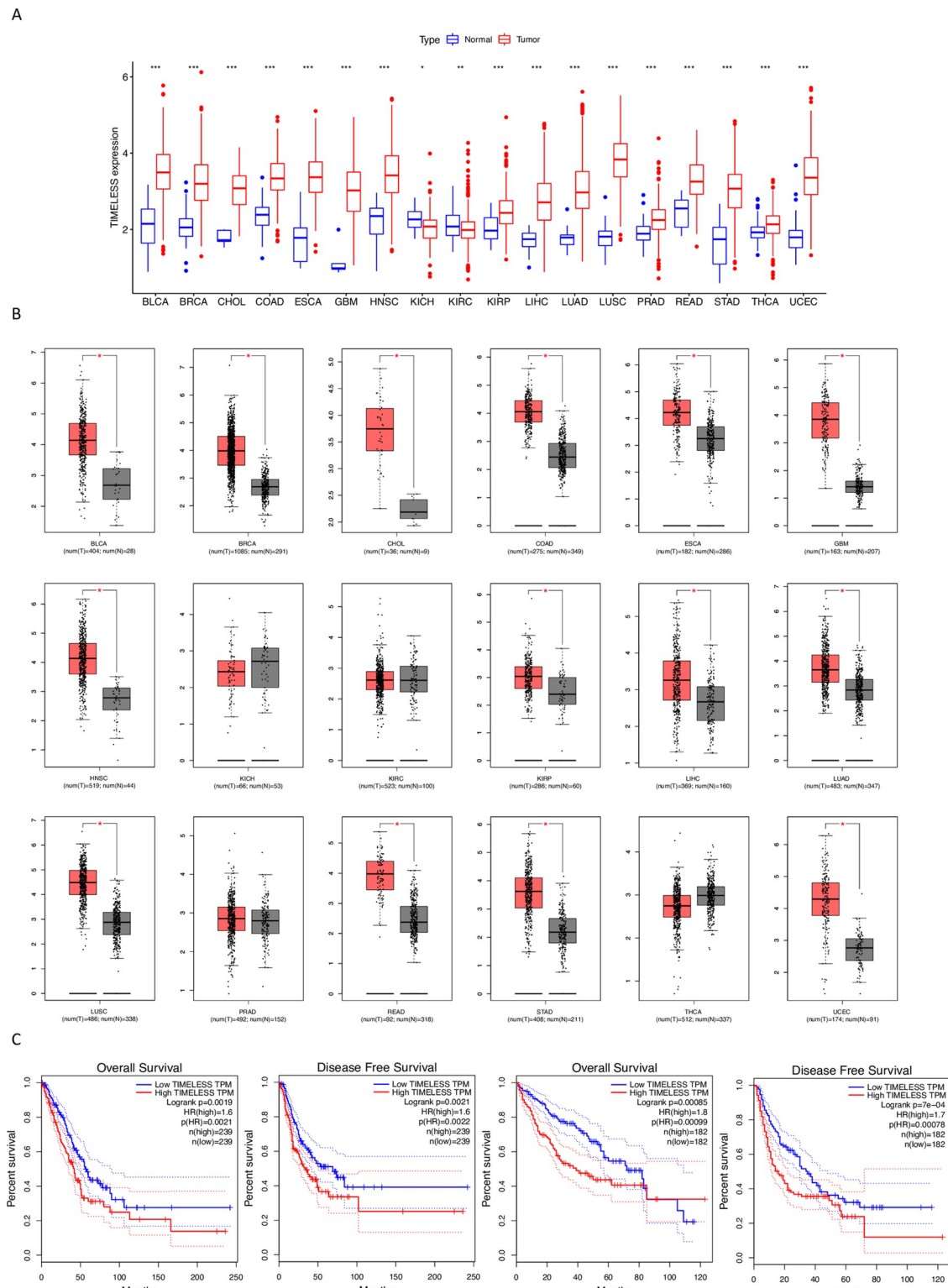

**Fig 1. *TIMELESS* expression and prognostic value across multiple types of cancer.** (A) *TIMELESS* expression in tumor tissues compared to adjacent normal tissues across 18 types of cancer in TCGA. (B) *TIMELESS* expression in TCGA tumor tissues (red) across18 types of cancer compared to normal tissues (grey) from TCGA and individuals without cancer from the GTEx Program. (C) The prognostic value of *TIMELESS* in LUAD (right two figures) and LIHC (left two figures). *p value < 0.05; **p value < 0.01; ***p value < 0.001.

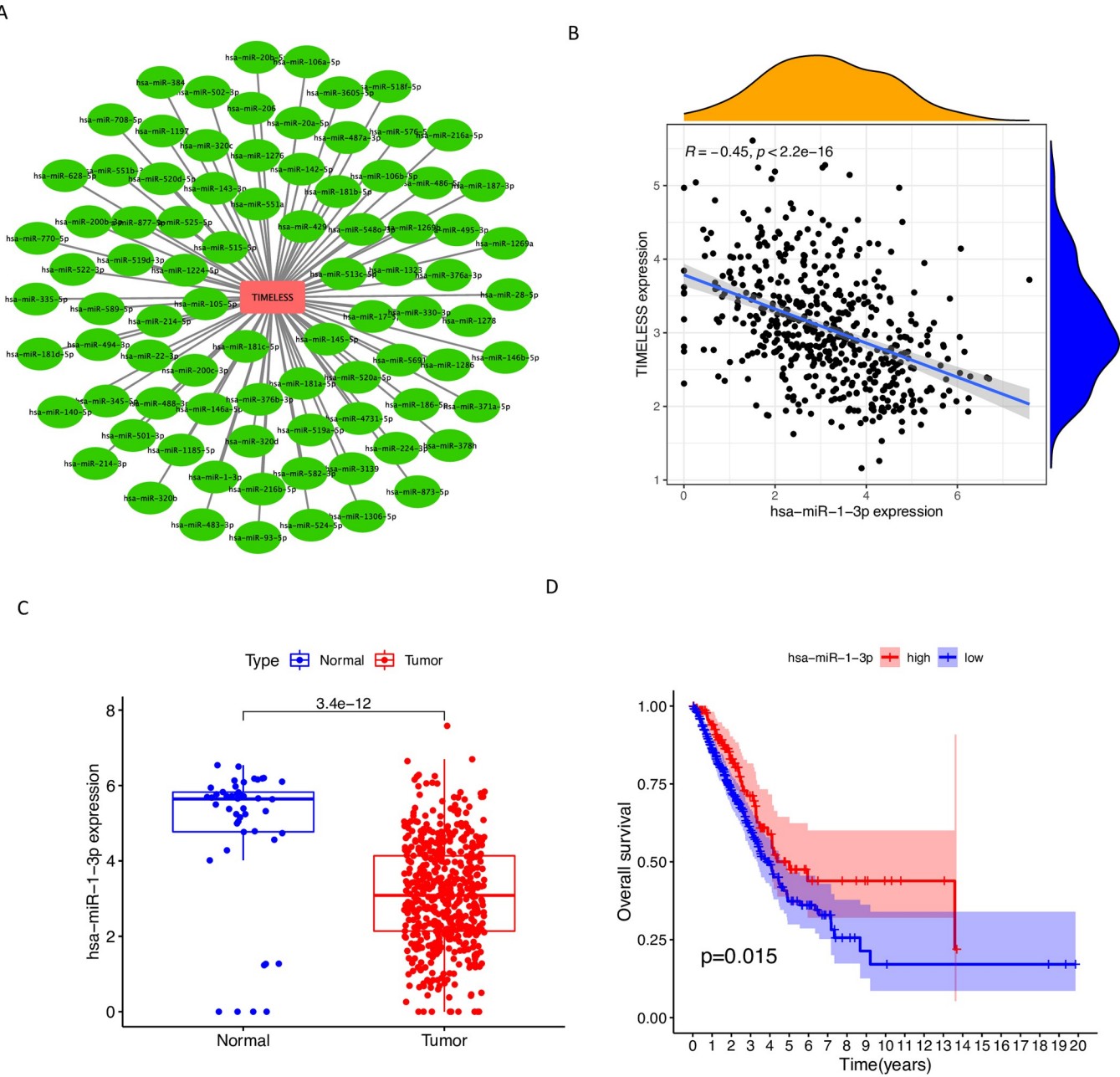

**Fig 2.** Identification of hsa-miR-1-3p as a potential upstream miRNA of *TIMELESS* in LUAD (A) The miRNA-*TIMELESS* interaction network (Cytoscape software). (B) Correlation between *TIMELESS* expression and expression of candidate miRNAs in TCGA LUAD tumor tissues (starBase database). (C) hsa-miR-1-3p expression in TCGA LUAD tumor tissues compared to adjacent normal tissues. (D) The prognostic value of hsa-miR-1-3p in LUAD.

## hsa-miR-1-3p upstream lncRNAs

A total of 92 lncRNAs with potential to interact with hsa-miR-1-3p were identified in TCGA LUAD tumor tissues. The lncRNAs most likely to participate in the lncRNA—hsa-miR-1-3p - *TIMELESS* interaction network in LUAD tumor tissue were selected according to the following criteria: significant upregulation of lncRNA expression in LUAD tumor tissues vs. adjacent normal tissues, significant negative correlation between lncRNA expression and hsa-miR-1-3p

**Table 1. Correlations between miRNA and *TIMELESS* expression in TCGA LUAD tumor tissues.**

| miRNA | R value | P value | logFC | P value |
|---|---|---|---|---|
| hsa-miR-1-3p | -0.452 | <0.001 | -1.685 | <0.001 |
| hsa-miR-145-5p | -0.265 | <0.001 | -0.492 | 0.308 |
| hsa-miR-181a-5p | -0.256 | <0.001 | -0.099 | 0.282 |
| hsa-miR-551a | -0.224 | <0.001 | 0.505 | <0.001 |
| hsa-miR-224-3p | -0.202 | <0.001 | 0.660 | <0.001 |

expression in LUAD tumor tissues, and significant association of high lncRNA expression in tumor tissues with poor prognosis in LUAD. Among the 92 lncRNAs with potential to interact with hsa-miR-1-3p, only *MIR4435-2HG*, *MIAT*, and *CYTOR (LINC00152)* were significantly upregulated in TCGA LUAD tumor tissues vs. adjacent normal tissues, with expression that significantly negatively correlated with hsa-miR-1-3p expression (**Fig 3A and 3B**). High *MIR4435-2HG* expression in tumor tissues was significantly associated with poor OS in the TCGA LUAD cohort (**Fig 3C**). Overexpressed *CYTOR* tended towards poor OS in LUAD but did not reach significance. These data suggest *MIR4435-2HG* and *CYTOR* have a potential role in the lncRNA—hsa-miR-1-3p - *TIMELESS* interaction network in LUAD (**Fig 3D**).

## Tumor immune cell infiltration and response to immune checkpoint inhibitors

*TIMELESS* plays a critical role in the immune system. ssGSEA scores for most immune cells were significantly decreased in TCGA LUAD tumor tissues with high *TIMELESS* expression vs. low *TIMELESS* expression. ssGSEA scores for activated CD4$^+$T cells and the immunosuppressive Th2 cells were significantly increased in TCGA LUAD tumor tissues with high *TIMELESS* expression vs. low *TIMELESS* expression (**Fig 4A**). A heatmap of immune cell infiltration confirmed that increased *TIMELESS* expression was associated with CD4$^+$ T and Th2 cell infiltration in TCGA LUAD tumor tissues (**Fig 4B**). *TIMELESS* expression was significantly negatively correlated with neutrophil biomarkers (*CEACAM8*) and dendritic cell biomarkers (*HLA-DPB1*, *HLA-DQB1*, *HLA-DRA*, *HLA-DPA1*, *CD1C*) in TCGA LUAD tumor tissues (**Table 2**). These data suggest *TIMELESS* is involved in immune regulation of the LUAD tumor microenvironment. *TIMELESS* expression was significantly negatively correlated with an immunophenoscore that predicted response to immune checkpoint inhibitors in TCGA LUAD tumor tissues (**Fig 5**). These data suggest patients with LUAD and high *TIMELESS* expression in their tumor tissue may be less responsive to immune checkpoint inhibitors than patients with lower *TIMELESS* expression in their tumor tissue.

## Discussion

This study used data from TCGA and the GTEx Program to explore *TIMELESS* expression and survival outcomes in human cancers. High *TIMELESS* expression in tumor tissue was significantly associated with poor OS in KIRC, LIHC, and LUAD and poor RFS in LIHC and LUAD. Consistent with this, other studies have shown a correlation between *TIMELESS* overexpression and poor prognosis in lung cancer [14, 15, 23], and *TIMELESS* has been used to build a predictive model of survival in lung cancer [24]. However, the regulation and mechanism of action of *TIMELESS* in lung cancer remains to be elucidated.

ncRNAs, including miRNAs and lncRNAs, are one component of post transcriptional regulation of gene expression. In the present study, *TIMELESS* expression was significantly negatively correlated with hsa-miR-1-3p expression in LUAD tumor tissues, and high hsa-miR-1-

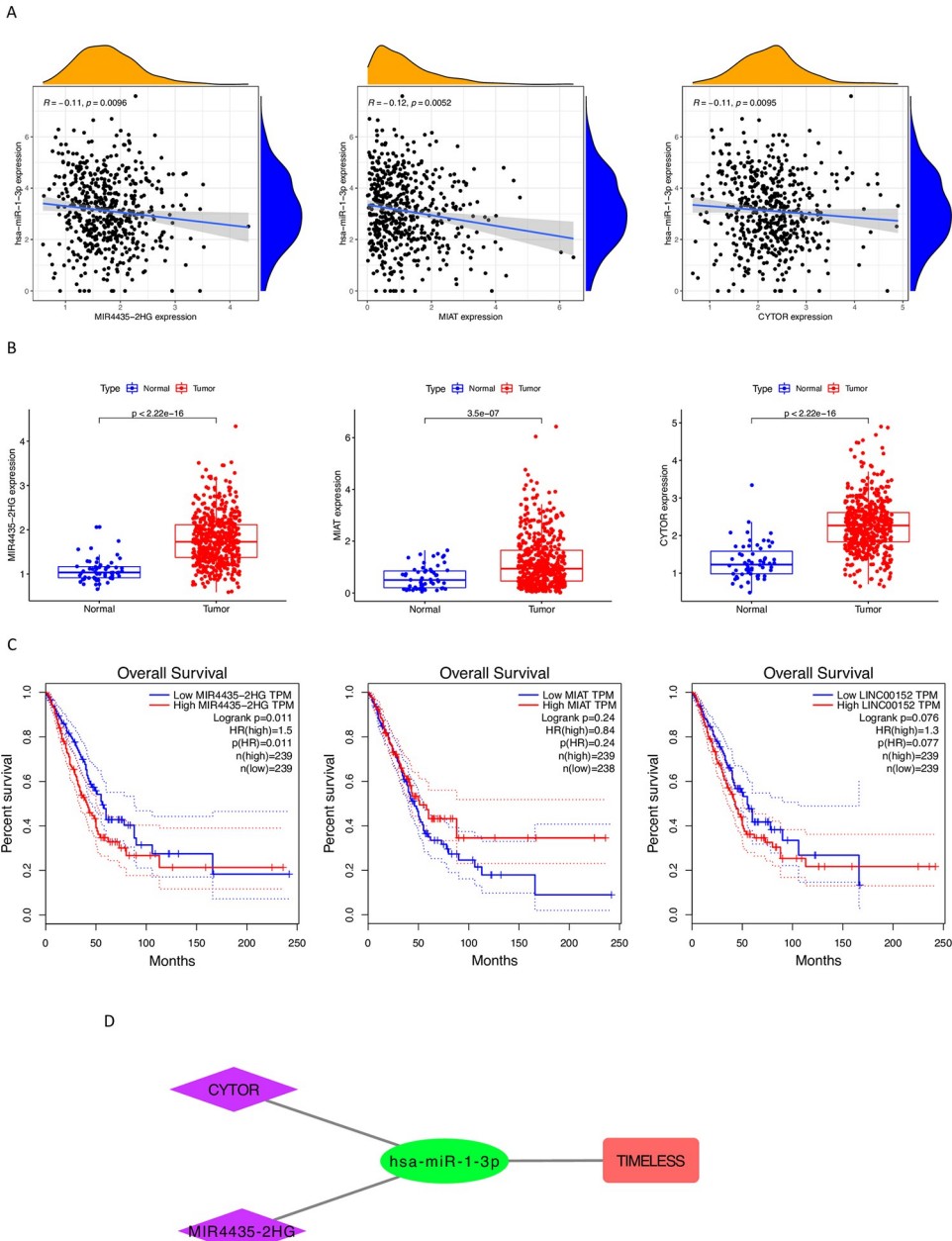

**Fig 3. Identification of potential upstream lncRNAs of hsa-miR-1-3p in LUAD.** (A) Correlation between hsa-miR-1-3p expression and the expression of candidate lncRNAs (MIR4435−2HG, MIAT, and CYTOR) in TCGA LUAD tumor tissues. (B) Candidate lncRNA (MIR4435−2HG, MIAT, and CYTOR) expression in TCGA LUAD tumor tissues compared to adjacent normal tissues. (C) The prognostic value of candidate lncRNAs (MIR4435−2HG, MIAT, and CYTOR) in LUAD. (D) The lncRNA—hsa-miR-1-3p - *TIMELESS* interaction network in LUAD tumor tissue.

3p expression was significantly associated with improved OS in LUAD. Previous reports indicate that miR-1-3p has an inhibitory role in lung cancer by targeting *CELSR3* and *FAM83A* and modulating the viability, migration, and invasion of tumor cells [25–27]. According to the ceRNA hypothesis [28], lncRNAs in the lncRNA—hsa-miR-1-3p - *TIMELESS* interaction network in LUAD should be oncogenic. Expression, survival and correlation analyses identified *MIR4435−2HG* as a candidate oncogenic lncRNA in the lncRNA—hsa-miR-1-3p - *TIMELESS*

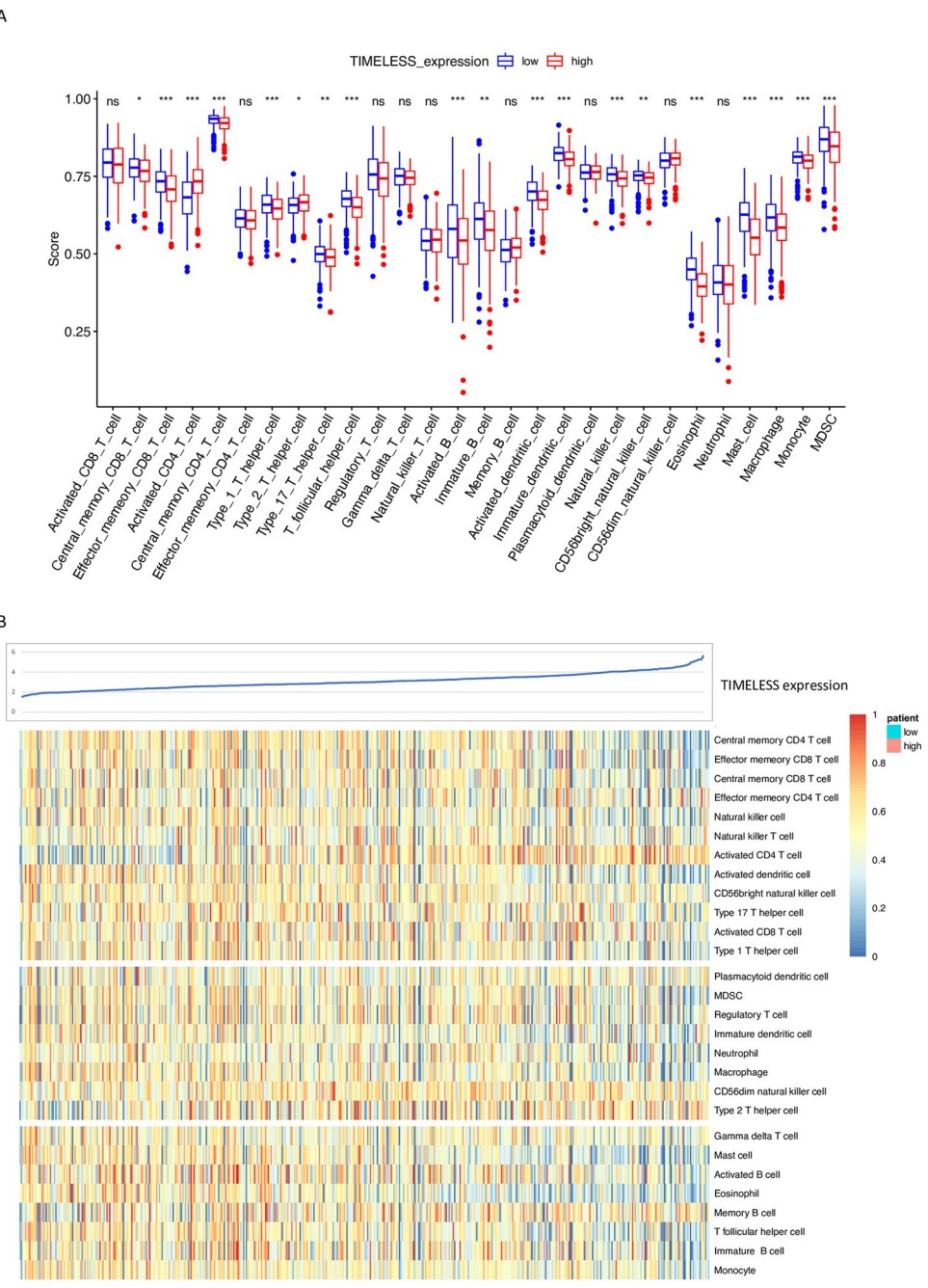

**Fig 4. Tumor immune cell infiltration and *TIMELESS* expression in LUAD.** (A) Correlation between *TIMELESS* expression and immune cell infiltration in TCGA LUAD tumor tissues. (B) Heatmap of immune cell infiltration in TCGA LUAD tumor tissues stratified by *TIMELESS* expression level. *p value < 0.05; **p value < 0.01; ***p value < 0.001.

interaction network in LUAD. Previous studies show lnc *MIR4435-2HG* may act as an onco-gene across several tumor types; specifically, *MIR4435–2HG* may promote lung cancer by acti-vating TGF-β1 and β-catenin signaling [29–32].

Research has revealed that some genes including circadian genes can alter tumor immune cell infiltration and influence prognosis [12, 33–35]. The present study adds to this evidence-

**Table 2. Correlations between *TIMELESS* expression and biomarkers of immune cells in TCGA LUAD tumor tissues.**

| Immune Cell | Gene | Spearman R | P value |
|---|---|---|---|
| B cell | CD19 | 0.011 | 0.804 |
| | CD79A | -0.022 | 0.612 |
| CD8+ T cell | CD8A | 0.080 | 0.066 |
| | CD8B | 0.112 | 0.010 |
| CD4+ T cell | CD4 | -0.188 | <0.001 |
| M1 macrophage | NOS2 | 0.075 | 0.086 |
| | IRF5 | -0.006 | 0.888 |
| | PTGS2 | 0.039 | 0.372 |
| M2 macrophage | CD163 | 0.002 | 0.959 |
| | VSIG4 | -0.103 | 0.018 |
| | MS4A4A | -0.172 | <0.001 |
| Neutrophil | CEACAM8 | -0.325 | <0.001 |
| | ITGAM | -0.161 | <0.001 |
| | CCR7 | -0.178 | <0.001 |
| Dendritic cell | HLA-DPB1 | -0.385 | <0.001 |
| | HLA-DQB1 | -0.283 | <0.001 |
| | HLA-DRA | -0.353 | <0.001 |
| | HLA-DPA1 | -0.316 | <0.001 |
| | CD1C | -0.531 | <0.001 |
| | NRP1 | -0.166 | <0.001 |
| | ITGAX | -0.020 | 0.639 |

base. High *TIMELESS* expression was associated with decreased infiltration of most immune cells in LUAD tumor tissues. Notably, high *TIMELESS* expression was associated with increased infiltration of activated CD4[+] T cells and immunosuppressive Th2 cells and decreased infiltration of CD8[+] T cells, cytotoxic Th1 cells, and dendritic cells in LUAD tumor tissues. These data imply *TIMELESS* is involved in immune regulation of the LUAD tumor microenvironment. Accordingly, a high percent of CD4[+] T cells in the tumor microenvironment may predict poor prognosis in NSCLC [36], and cytotoxic Th1 cells and dendritic cells

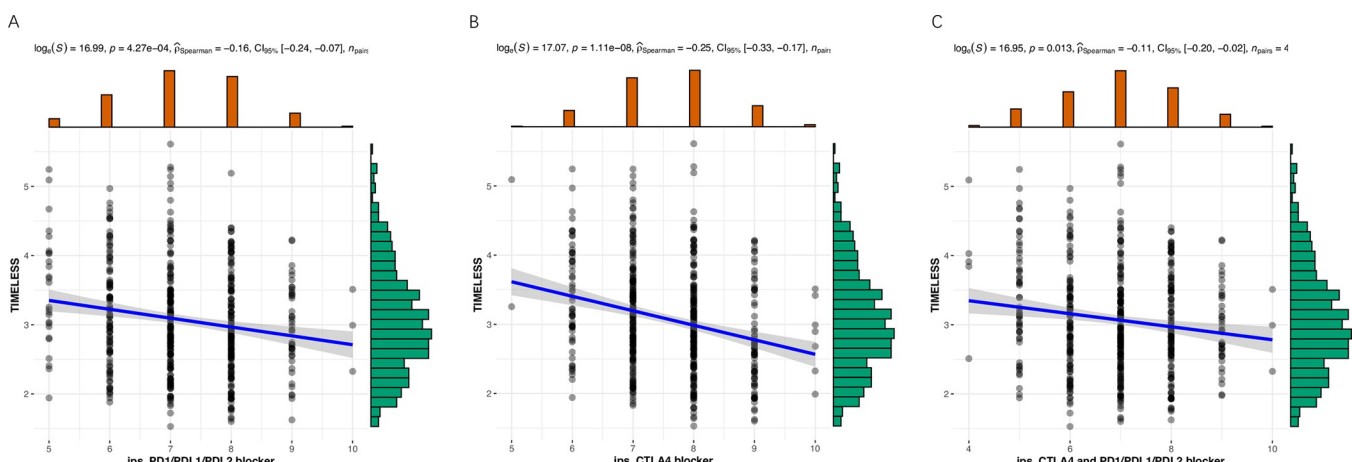

**Fig 5. Correlation of *TIMELESS* expression with immunophenoscores that predict response to immune checkpoint inhibitors in TCGA LUAD tumor tissues.**

can maintain anti-tumor immunity [36]. Further, *TIMELESS* expression in LUAD tumor tissues was significantly negatively correlated with an immunophenoscore that predicted response to immune checkpoint inhibitors [22], reflecting the impact of *TIMELESS* on immune cell infiltration. As is a circadian gene, immune checkpoint inhibitor regimens may need to be optimally timed with a patient's intrinsic rhythms to improve outcomes. Currently, only chronoradiotherapy and chronochemotherapy have been investigated across several cancers [37–39].

This study had some limitations. First, data were extracted from publicly available databases. Second, immunity-related analyses were based on bioinformatics, with no physiological validation. Third, lung cancer is associated with multiple molecular abnormalities, but we only focused on the lncRNA—miRNA—interaction network, potentially excluding other critical genes.

## Conclusions

In summary, this study identified *TIMELESS* as a potential oncogene across multiple cancers, including LUAD. High *TIMELESS* expression in tumor tissues was significantly associated with poor OS and RFS in LUAD. *MIR4435-2HG* and hsa-miR-1-3p may interact with *TIMELESS* in a ceRNA network in LUAD tumor tissues. *TIMELESS* might exert its oncogenic role by decreasing immune cell infiltration in the LUAD tumor microenvironment, and *TIMELESS* expression in LUAD tumor tissues may predict a poor response to immune checkpoint inhibitors. Basic science and clinical trials are required to validate these findings.

## Author Contributions

**Conceptualization:** Xinliang Gao, Suyan Tian, Wei Liu.

**Data curation:** Xinliang Gao, Mingbo Tang, Shixiong Wei.

**Formal analysis:** Xinliang Gao, Jialin Li.

**Funding acquisition:** Shucheng Hua, Wei Liu.

**Methodology:** Xinliang Gao, Suyan Tian.

**Software:** Xinliang Gao, Mingbo Tang.

**Supervision:** Shucheng Hua, Wei Liu.

**Writing – original draft:** Xinliang Gao.

**Writing – review & editing:** Mingbo Tang, Suyan Tian, Jialin Li, Shixiong Wei, Wei Liu.

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
