## [Decision Letter · Decision Letter 0]

20 Sep 2023

PONE-D-23-17732ncRNAs-mediated TIMELESS overexpression  in lung adenocarcinoma correlates with reduced tumor immune cell infiltration and poor prognosisPLOS ONE

Dear Dr. Liu,

Thank you for submitting your manuscript to PLOS ONE. After careful consideration, we feel that it has merit but does not fully meet PLOS ONE’s publication criteria as it currently stands. Therefore, we invite you to submit a revised version of the manuscript that addresses the points raised during the review process.

We look forward to receiving your revised manuscript.

Kind regards,

Yuanliang Yan

Academic Editor

PLOS ONE

Journal Requirements:

a) The name of the colleague or the details of the professional service that edited your manuscript.

b) A copy of your manuscript showing your changes by either highlighting them or using track changes (uploaded as a *supporting information* file).

c) A clean copy of the edited manuscript (uploaded as the new *manuscript* file).

3. Thank you for stating the following financial disclosure: "No"

4. Thank you for stating the following in your Competing Interests section: "NO authors have competing interests"

Reviewers' comments:

Reviewer's Responses to Questions

**Comments to the Author**

1. Is the manuscript technically sound, and do the data support the conclusions?

Reviewer #1: Yes

Reviewer #2: Partly

2. Has the statistical analysis been performed appropriately and rigorously? 

Reviewer #1: Yes

Reviewer #2: I Don't Know

3. Have the authors made all data underlying the findings in their manuscript fully available?

Reviewer #1: Yes

Reviewer #2: No

4. Is the manuscript presented in an intelligible fashion and written in standard English?

Reviewer #1: Yes

Reviewer #2: Yes

5. Review Comments to the Author

Reviewer #1: The article discusses the correlation between non-coding RNAs (ncRNAs)-mediated TIMELESS overexpression in lung adenocarcinoma (LUAD) and poor prognosis, reduced immune cell infiltration in the tumor microenvironment, and poor response to immune checkpoint inhibitors. The study investigated the expression of TIMELESS, a circadian gene, and its association with survival in several types of human cancer using data from The Cancer Genome Atlas (TCGA) and the Genotype-Tissue Expression Program. The article also explores how MIR4435-2HG/hsa-miR-1-3p ncRNAs may interact with TIMELESS in a competitive endogenous RNA network in LUAD tumor tissues. However, some limitations are still existing.

1. The software used in the article, such as the software needed for drawing and data analysis, is not written in the method part. It is suggested to add it to the appropriate position in this part.

2. Generally, the spearman correlation coefficient |R| is set to >0.3 or 0.5, but what is the reason for setting the thresholds |R| >0.2 and |R| >0.1 in the Candidate miRNA and lncRNA prediction section? It is recommended to explain clearly.

3. In addition, there are still some details in the article that need to be improved, whether the R value and p-value in Table 1 and Table 2 can take the same number of decimals. Figure 1B does not have a legend indicating which is the cancer group and which is the control group. LIHC appears in the notes of Figure 1C, but the text results section (including the Discussion section) is written as KIHC, please check and revise to ensure accuracy and consistency. What cancers do the first pictures to the fourth pictures in Figure 1C correspond to respectively? It is suggested to indicate them in the figure notes to avoid misunderstanding.

4. In the discussion, the authors should discuss other genes and lung adenocarcinoma and cite the follows:

Yu Y, Wang Z, Zheng Q, Li J. GREB1L overexpression correlates with prognosis and immune cell infiltration in lung adenocarcinoma. Sci Rep. 2021 Jun 24;11(1):13281. doi: 10.1038/s41598-021-92695-x. PMID: 34168239; PMCID: PMC8225624.

Bai R, Zhang J, He F, Li Y, Dai P, Huang Z, Han L, Wang Z, Gong Y, Xie C. GPR87 promotes tumor cell invasion and mediates the immunogenomic landscape of lung adenocarcinoma. Commun Biol. 2022 Jul 5;5(1):663. doi: 10.1038/s42003-022-03506-6. PMID: 35790819; PMCID: PMC9256611.

Qiu A, Xu H, Mao L, Xu B, Fu X, Cheng J, Zhao R, Cheng Z, Liu X, Xu J, Zhou Y, Dong Y, Tian T, Tian G, Chu M. A Novel apaQTL-SNP for the Modification of Non-Small-Cell Lung Cancer Susceptibility across Histological Subtypes. Cancers (Basel). 2022 Oct 28;14(21):5309. doi: 10.3390/cancers14215309. PMID: 36358727; PMCID: PMC9658938.

Reviewer #2: Dear Authors:

You meticulously made use of several Open databases including TCGA, and attended To make a conclusion as in its abstract. “ that ncRNAs-mediated TIMELESS overexpression in LUAD tumor tissues correlated with poor prognosis, reduced immune cell infiltration in the tumor microenvironment, and poor response to immune checkpoint inhibitors.”

However, from our angle, there are still several conceptual gaps from their works and the augment they want to make.

First, the negative. correlation disclosed on Figure 5 is crucial to this conclusion. The also could have provided more Evidence of its statistical significance.

Second, you repeatedly emphasized the overexpression of TIMELESS is negatively correlated with hsa-miR-1-3p, which has well-known inhibitory role in Lung adenocarcino-ma, and they really proved it. The authors should have made Some efforts to Address the possibility of coincidence.

Third, the study design is also a little confusing. The authors stated” we selected 18 13 tumor types that contained more than five normal tissue samples (BLCA, BRCA, CHOL, COAD, ESCA, 14 GBM, HNSC, KICH, KIRC, KIRP, LIHC, LUAD, LUSC, PRAD, READ, STAD, THCA, and UCEC).” Why you finally Only reported findings of LUAD . Do other Associated findings not significant ?

Finally, too many abbreviations without clear explanation have been all over the context (as the cancer entities we listed above) . Even we are familiar with the terms as OS and PRS , you should have well revised the “List of abbreviations” on page to the best of readers’ convenience.

6. PLOS authors have the option to publish the peer review history of their article (what does this mean?). If published, this will include your full peer review and any attached files.

Reviewer #1: No

Reviewer #2: No

---

## [Author Response · Author response to Decision Letter 0]

5 Oct 2023

Dear editor and reviewers,

Thank you very much for your comments on our manuscript entitled “ncRNAs-mediated TIMELESS overexpression in lung adenocarcinoma correlates with reduced tumor immune cell infiltration and poor prognosis”. We have revised the manuscript accordingly, and all new amendments are indicated by the red font in the revised manuscript. In addition, our point-by-point responses to the comments are listed below this letter.

We hope that our revised manuscript could be acceptable for publication in your journal and look forward to hearing from you soon. 

With best wishes,

Yours sincerely,

Wei Liu

 

Reviewer 1 

1. The software used in the article, such as the software needed for drawing and data analysis, is not written in the method part. It is suggested to add it to the appropriate position in this part.

Answer: Thank you for your feedback regarding the inclusion of software details in the Method section of our article. We appreciate your suggestion and have addressed it by adding the necessary information to the appropriate position, which is marked in red.

2. Generally, the spearman correlation coefficient |R| is set to >0.3 or 0.5, but what is the reason for setting the thresholds |R| >0.2 and |R| >0.1 in the Candidate miRNA and lncRNA prediction section? It is recommended to explain clearly.

Answer: Thank you for your valuable feedback on our article regarding the Spearman correlation coefficients used in the Candidate miRNA and lncRNA prediction section. We appreciate your thoughtful comments. In this article, we used a public database in which miRNA and lncRNA prediction might have exhibited lower correlation coefficients due to inherent noise, biological complexity, or multiple regulatory mechanisms. A stricter threshold, such as |R| > 0.3 or 0.5, could have resulted in an undue exclusion of potentially relevant relationships for later wet experiment validation. The choice of |R| > 0.2 and |R| > 0.1 as correlation thresholds in our study was a deliberate decision to suit our research's specific characteristics and objectives. We believe these thresholds allowed us to capture meaningful associations in our dataset without prematurely excluding potentially valuable information. We hope this explanation clarifies our rationale for selecting these thresholds, and we are open to further discussion or adjustments if deemed necessary.

3. In addition, there are still some details in the article that need to be improved, whether the R value and p-value in Table 1 and Table 2 can take the same number of decimals. Figure 1B does not have a legend indicating which is the cancer group and which is the control group. LIHC appears in the notes of Figure 1C, but the text results section (including the Discussion section) is written as KIHC, please check and revise to ensure accuracy and consistency. What cancers do the first pictures to the fourth pictures in Figure 1C correspond to respectively? It is suggested to indicate them in the figure notes to avoid misunderstanding.

Answer: We are grateful for your thorough review of our article and for providing valuable suggestions. We have taken your comments seriously and made the necessary revisions to enhance the clarity and accuracy of our article. 1) We have ensured that the R values and p-values in Table 1 and Table 2 are consistent, displaying three decimal places for accuracy and clarity. 2) We have added clear labels in the legend of Figure 1B to distinguish between the cancer group and the control group, thereby enhancing the interpretability of the figure. 3) Sorry for making the misunderstanding of writing LICH as KICH. We have diligently corrected all instances of "KIHC" to "LIHC" in both the results and discussion sections to maintain accuracy throughout the article. 4) We have labeled the cancers corresponding to the first to fourth pictures in Figure 1C. Additionally, we have indicated these labels in the figure legend to eliminate any potential misunderstandings.

4. In the discussion, the authors should discuss other genes and lung adenocarcinoma and cite the follows:

Yu Y, Wang Z, Zheng Q, Li J. GREB1L overexpression correlates with prognosis and immune cell infiltration in lung adenocarcinoma. Sci Rep. 2021 Jun 24;11(1):13281. doi: 10.1038/s41598-021-92695-x. PMID: 34168239; PMCID: PMC8225624.

Bai R, Zhang J, He F, Li Y, Dai P, Huang Z, Han L, Wang Z, Gong Y, Xie C. GPR87 promotes tumor cell invasion and mediates the immunogenomic landscape of lung adenocarcinoma. Commun Biol. 2022 Jul 5;5(1):663. doi: 10.1038/s42003-022-03506-6. PMID: 35790819; PMCID: PMC9256611.

Qiu A, Xu H, Mao L, Xu B, Fu X, Cheng J, Zhao R, Cheng Z, Liu X, Xu J, Zhou Y, Dong Y, Tian T, Tian G, Chu M. A Novel apaQTL-SNP for the Modification of Non-Small-Cell Lung Cancer Susceptibility across Histological Subtypes. Cancers (Basel). 2022 Oct 28;14(21):5309. doi: 10.3390/cancers14215309. PMID: 36358727; PMCID: PMC9658938.

Answer: We appreciate your feedback and your valuable suggestions for improving the depth and comprehensiveness of our discussion in the article. We have carefully reviewed your comments and have made the necessary additions to the manuscript by incorporating the suggested articles and highlighting them in red. Our aim is to enhance the discussion by considering additional genes and their relevance to lung adenocarcinoma, as per your recommendation.

Reviewer 2

1. First, the negative correlation disclosed on Figure 5 is crucial to this conclusion. The also could have provided more Evidence of its statistical significance.

Answer: Thank you for your insightful comment regarding the statistical significance of the negative correlation presented in Figure 5. We appreciate your interest in our findings' robustness and would like to address your concern. In our study, we have also explored the analysis of the TIMELESS gene with immune cell infiltration, as demonstrated in Figure 4. This analysis reveals that the degree of infiltration of multiple immune cells is reduced in the TIMELESS high-expression group. This additional analysis further supports the hypothesis that TIMELESS may be negatively correlated with the efficacy of immunotherapy from an alternative perspective. We acknowledge that the most direct and definitive evidence for this hypothesis would involve comparing TIMELESS expression in patients who respond differently to immunotherapy. However, it is essential to clarify that the TCGA database we utilized for this study does not provide the complete clinical sample information required to conduct such a comparison. To address this limitation, we are actively collecting and sorting relevant clinical samples, and we are in the process of obtaining more comprehensive clinical data from our own center. We are committed to conducting an in-depth analysis that directly compares TIMELESS expression in patients with varying responses to immunotherapy. The results of this analysis will be included in our subsequent research articles, providing the robust clinical evidence you rightly emphasized.

2. Second, you repeatedly emphasized the overexpression of TIMELESS is negatively correlated with hsa-miR-1-3p, which has well-known inhibitory role in Lung adenocarcinoma, and they really proved it. The authors should have made Some efforts to Address the possibility of coincidence.

Answer: Thank you for your insightful comment regarding the potential for coincidence in our observation of the negative correlation between TIMELESS and hsa-miR-1-3p in Lung Adenocarcinoma (LUAD). We appreciate your concern and would like to address it comprehensively. You are correct in emphasizing that bioinformatics analysis alone cannot establish a causal relationship between miRNA and target gene interactions. We completely agree that the correlation observed in our study needed validation through experimental approaches to confirm the functional relevance. We want to inform you that we have conducted relevant experimental investigations to address this concern. Specifically, we performed experiments involving two cell lines, A549 and H1299, which naturally low-express miR-1-3p. In these experiments, we observed a down-regulation of TIMELESS expression after up-regulating miR-1-3p. Furthermore, our findings were supported by luciferase reporter assays, which demonstrated binding interactions between miR-1-3p and TIMELESS within the cells. We are pleased to share that the results of these experimental validations have been prepared as a separate research article and are currently under peer review. We are committed to providing conclusive evidence for the functional relationship between hsa-miR-1-3p and TIMELESS in LUAD. We apologize for not including these experimental findings in the current article, but assure you they are being actively pursued and will be published in due course.

3. Third, the study design is also a little confusing. The authors stated” we selected 18 13 tumor types that contained more than five normal tissue samples (BLCA, BRCA, CHOL, COAD, ESCA, 14 GBM, HNSC, KICH, KIRC, KIRP, LIHC, LUAD, LUSC, PRAD, READ, STAD, THCA, and UCEC).” Why you finally Only reported findings of LUAD. Do other Associated findings not significant?

Answer: We appreciate your thoughtful inquiry regarding the study design and the selection of specific tumor types for reporting in our article. We want to clarify the rationale behind our choices. Our research initially screened 14 tumor types that exhibited differential TIMELESS gene expression between tumor and normal tissues based on comprehensive data from the TCGA and GTEx databases. This initial screening was essential to identify potential associations between TIMELESS expression and cancer. Following this screening, we narrowed our focus to LUAD (Lung Adenocarcinoma) based on the following considerations: 1) We further investigated the prognostic implications of TIMELESS expression within these 14 tumor types. Specifically, we examined the survival outcomes of patients with high versus low TIMELESS expression in each tumor type. LUAD and LIHC were the two tumor types where TIMELESS expression was negatively associated with patient prognosis. This observation led us to select them for further analysis. 2) Additionally, our research team possesses specialized expertise in thoracic surgery, which made LUAD a particularly relevant and compelling choice for in-depth analysis. This expertise allowed us to offer more comprehensive insights into the clinical implications of TIMELESS expression in the context of LUAD.

4. Finally, too many abbreviations without clear explanation have been all over the context (as the cancer entities we listed above) . Even we are familiar with the terms as OS and PRS , you should have well revised the “List of abbreviations” on page to the best of readers’ convenience.

Answer: We sincerely appreciate your feedback regarding the use of abbreviations in our article. We recognize the importance of clarity and reader convenience and have taken your comment to heart. To address this concern, we have made significant revisions to the manuscript. Specifically, we have expanded the full names of abbreviations wherever they first appear in the text to ensure readers can easily understand the context. We believe these revisions will significantly enhance the readability and comprehensibility of our manuscript.

---

## [Decision Letter · Decision Letter 1]

20 Dec 2023

ncRNAs-mediated TIMELESS overexpression  in lung adenocarcinoma correlates with reduced tumor immune cell infiltration and poor prognosis

PONE-D-23-17732R1

Dear Dr. Liu,

We’re pleased to inform you that your manuscript has been judged scientifically suitable for publication and will be formally accepted for publication once it meets all outstanding technical requirements.

Kind regards,

Yuanliang Yan

Academic Editor

PLOS ONE

Additional Editor Comments (optional):

Reviewers' comments:

Reviewer's Responses to Questions

**Comments to the Author**

1. If the authors have adequately addressed your comments raised in a previous round of review and you feel that this manuscript is now acceptable for publication, you may indicate that here to bypass the “Comments to the Author” section, enter your conflict of interest statement in the “Confidential to Editor” section, and submit your "Accept" recommendation.

Reviewer #1: All comments have been addressed

2. Is the manuscript technically sound, and do the data support the conclusions?

Reviewer #1: Yes

3. Has the statistical analysis been performed appropriately and rigorously? 

Reviewer #1: Yes

4. Have the authors made all data underlying the findings in their manuscript fully available?

Reviewer #1: Yes

5. Is the manuscript presented in an intelligible fashion and written in standard English?

Reviewer #1: Yes

6. Review Comments to the Author

Reviewer #1: Thanks for the reply of the questions. I have no further questions, and this paper can be accepted with no revision.

7. PLOS authors have the option to publish the peer review history of their article (what does this mean?). If published, this will include your full peer review and any attached files.

Reviewer #1: No

---

## [Editor Report · Acceptance letter]

15 Jan 2024

PONE-D-23-17732R1 

PLOS ONE

Dear Dr. Liu, 

I'm pleased to inform you that your manuscript has been deemed suitable for publication in PLOS ONE. Congratulations! Your manuscript is now being handed over to our production team.

Kind regards, 

on behalf of

Prof. Yuanliang Yan 

Academic Editor

PLOS ONE